# PartEdit: Fine-Grained Image Editing using Pre-Trained Diffusion Models

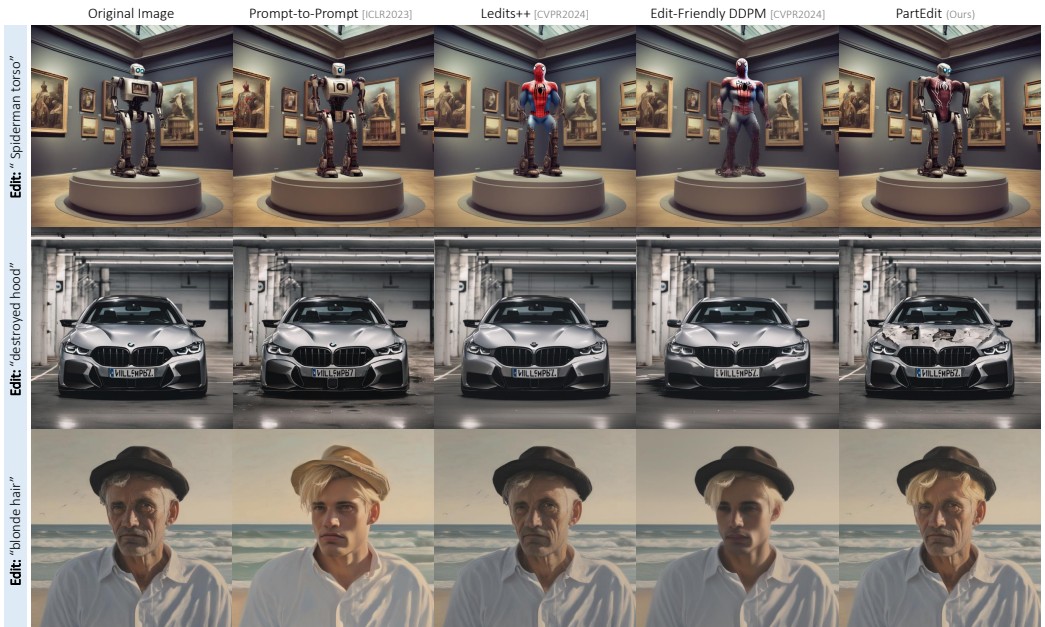

Figure 1: Our approach, *PartEdit*, enables a wide range of fine-grained edits, allowing users to create highly customizable changes. The edits are seamless, precisely localized, and of high visual quality with no leakage into unedited regions.

## ABSTRACT

We present the first *text-based* image editing approach for *object parts* based on pre-trained diffusion models. Diffusion-based image editing approaches capitalized on the deep understanding of diffusion models of image semantics to perform a variety of edits. However, existing diffusion models lack sufficient understanding of many object parts, hindering fine-grained edits requested by users. To address this, we propose to expand the knowledge of pre-trained diffusion models to allow them to understand various object parts, enabling them to perform fine-grained edits. We achieve this by learning special textual tokens that correspond to different object parts through an efficient token optimization process. These tokens are optimized to produce reliable localization masks *at each inference step* to localize the editing region. Leveraging these masks, we design feature-blending and adaptive thresholding strategies to execute the edits seamlessly. To evaluate our approach, we establish a benchmark and an evaluation protocol for part editing. Experiments show that our approach outperforms existing editing methods on all metrics and is preferred by users $77 - 90\%$ of the time in conducted user studies.

**Prompt:** *"A muscular man in a white shirt with a robotic **head**"*  **Prompt:** *"A red mustang car with a black **hood** parked by the ocean"*

Figure 2: A visualization for the cross-attention maps of SDXL (Podell et al., 2024) that corresponds to different words of the textual prompt. Object parts such as "head" and "hood" are not well-localized, indicating that the model lacks a sufficient understanding of these parts.

## 1 INTRODUCTION

Diffusion models (Rombach et al., 2022; Ramesh et al., 2022; Saharia et al., 2022; Podell et al., 2024; Esser et al., 2024) have significantly advanced image generation, achieving unprecedented levels of quality and fidelity. This progress is generally attributed to their large-scale training on the LAION-5B dataset (Schuhmann et al., 2022) with image-text pairs, leading to a profound understanding of images and their semantics. Recent image editing methods (Hertz et al., 2022; Brooks et al., 2023; Brack et al., 2024; Parmar et al., 2023; Tumanyan et al., 2023; Kawar et al., 2023; Huang et al., 2024) have capitalized on this understanding to perform a wide range of edits to enhance the creative capabilities of artists and designers. These methods allow users to specify desired edits through text prompts, enabling both semantic edits, such as modifying objects or their surroundings, and artistic adjustments, like changing style and texture. Ideally, these edits must align with the requested textual prompts while being seamlessly integrated and accurately localized in the image.

Despite the remarkable advancement in these diffusion-based image editing methods, their effectiveness is limited by the extent to which diffusion models understand images. For instance, while a diffusion model can manipulate objects, it might fail to perform edits on fine-grained object parts. Figure 1 shows examples where existing editing methods fail to perform fine-grained edits. For instance, in the first row, they exhibit *poor localization* of the editing region and fail to edit *only* the torso of the robot. In the second row, none of the approaches were able to localize the "hood" or apply the edit. In the final row, existing approaches suffer from *entangled parts*, making the man's face younger when instructed to change his hair to "blonde." These limitations can be attributed to the coarse textual descriptions in the LAIOB-5B dataset that is used to train diffusion models. Specifically, the model fails to understand various object parts as they are not explicitly described in image descriptions. Moreover, data biases in the datasets are also captured by the model, *e.g.*, associating blond hair with youth.

To validate this hypothesis, we visualized the cross-attention maps of a pre-trained diffusion model for two textual prompts with specific object parts in Figure 2. Cross-attention computes attention between image features and textual tokens, reflecting where each word in the textual prompt is represented in the generated image. For the first prompt, *"A muscular man in a white shirt with a robotic head"*, the generated image resembles a man with robotic arms instead of a robotic head. The cross-attention maps show that the token "head" is activated at the arms rather than the head. A possible explanation for this behavior is that the "head" has been entangled with the "arms" during training due to the coarse textual annotations of the training images. For the second prompt, *"A red Mustang car with a black hood parked by the ocean"*, the generated car is entirely red, including the hood. The cross-attention map for the token "hood" indicates that the model is uncertain about its location. This can be attributed to the lack or scarcity of images with "hood" annotations in the LAION-5B dataset used for training.

In this paper, we address the limitations of pre-trained diffusion models in their understanding of object parts. By expanding their semantic knowledge, we enable fine-grained image edits, providing creators with greater control over their images. We achieve this by training part-specific tokens that specialize in localizing the editing region at each denoising step. Based on this localization, we develop feature blending and adaptive thresholding strategies that ensure seamless and high-quality editing while preserving the unedited areas. To learn the part tokens, we design a token optimization process (Zhou et al., 2022) tailored for fine-grained editing, utilizing existing object part

---
*Equal contribution

datasets (Donadello & Serafini, 2016; He et al., 2022) or user-provided datasets. This optimization process allows us to keep the pre-trained diffusion model frozen, thereby expanding its semantic understanding without compromising generation quality or existing knowledge. To evaluate our approach and to facilitate the development of future fine-grained editing approaches, we introduce a benchmark and an evaluation protocol for part editing. Experiments show that our approach outperforms all methods in comparison on all metrics and is preferred by users $77 - 90\%$ of the time in conducted user studies.

## 2 TEXT-TO-IMAGE DIFFUSION MODELS

Diffusion models are probabilistic generative models that attempt to learn an approximation of a data distribution $p(\mathbf{x})$. This is achieved by progressively adding noise to each data sample $\mathbf{x}_0 \sim p(\mathbf{x})$ throughout $T$ timesteps until it converges to an isotropic Gaussian distribution as $T \rightarrow \infty$. During inference, a sampler such as DDIM (Song et al., 2020) is used to reverse this process starting from Gaussian noise $\mathbf{x}_T \sim \mathcal{N}(\mathbf{0}, \mathbf{I})$ that is iteratively denoised until we obtain a noise-free sample $\hat{\mathbf{x}}_0^0$. The reverse process for timestep $t \in [T, 0]$ is computed as:

$$x_{t-1} = \sqrt{\alpha_{t-1}}\, \hat{x}_0^t + \sqrt{1 - \alpha_{t-1} - \sigma_t^2}\, \epsilon_\theta^t(x_t) + \sigma_t \epsilon_t \;\;,$$
$$\hat{x}_0^t = \frac{x_t - \sqrt{1 - \alpha_t}\, \epsilon_\theta^t(x_t)}{\sqrt{\alpha_t}} \;. \tag{1}$$

where $\alpha_t, \sigma_t$ are scheduling parameters, $\epsilon_\theta^t$ is a noise prediction from the UNet, and $\epsilon_t$ is random Gaussian noise. This is referred to as unconditional sampling, where the model would generate an arbitrary image for every random initial noise $x_T$.

To generate an image that adheres to a user-provided input, *e.g.*, a textual prompt $\mathcal{P}$, the model can be trained conditionally. In this setting, the UNet is conditioned on the prompt $\mathcal{P}$, and the noise prediction in Equation (1) is computed as $\epsilon_\theta^t(x_t, \mathcal{P})$. More specifically, the textual prompt is embedded through a textual encoder, *e.g.*, CLIP (Radford et al., 2021), to obtain an embedding $E \in \mathrm{R}^{77 \times 2048}$ (for SDXL). This textual embedding interacts with the UNet image features $F$ within cross-attention modules at UNet block $i$ to compute attention as:

$$A_i = \mathrm{Softmax}\left(\frac{Q_i\, K_i^\top}{\sqrt{d_{k_i}}}\right) \;\;,$$
$$Q_i = F_i\, W_{Q_i}, \qquad K_i = E\, W_{K_i}, \qquad V_i = E\, W_{V_i}. \tag{2}$$

where $F_i$ are UNet features at layer $i$, and $W$ are trainable projection matrices. The output features are eventually recomputed as $\hat{F}_i = A_i\, V_i$. This interaction between the text embedding $E$ and the image features $F$ allows cross-attention modules to capture how each word/token in the text prompt spatially contributes to the generated image as illustrated in Figure 2. Similarly, each UNet block has a self-attention module that computes cross-feature similarities encoding the style of the generated image where:

$$Q_i = F_i\, W_{Q_i}, \qquad K_i = F_i\, W_{K_i}, \qquad V_i = F_i\, W_{V_i}.$$

## 3 PARTEDIT: FINE-GRAINED IMAGE EDITING

A common approach for editing images in diffusion-based methods involves manipulating the cross-attention maps (Hertz et al., 2022; Parmar et al., 2023; Epstein et al., 2023). If the cross-attention maps do not accurately capture the editing context, *e.g.*, for editing object parts, these methods are likely to fail, as demonstrated in Figure 1. An intuitive solution to this problem is expanding the knowledge of the pre-trained diffusion models to understand object parts. This can be accomplished by fine-tuning the model with additional data of image/text pairs where the text is detailed. However, this approach can be costly due to the extensive annotation required for fine-tuning the model, and there is no guarantee that the model will automatically learn to identify object parts effectively from text.

An alternative approach is leveraging token optimization (Zhou et al., 2022) to learn new concepts through explicit supervision of cross-attention maps. This was proven successful in several applications (Hedlin et al., 2023; Khani et al., 2024; Marcos-Manchón et al., 2024) as it allows learning

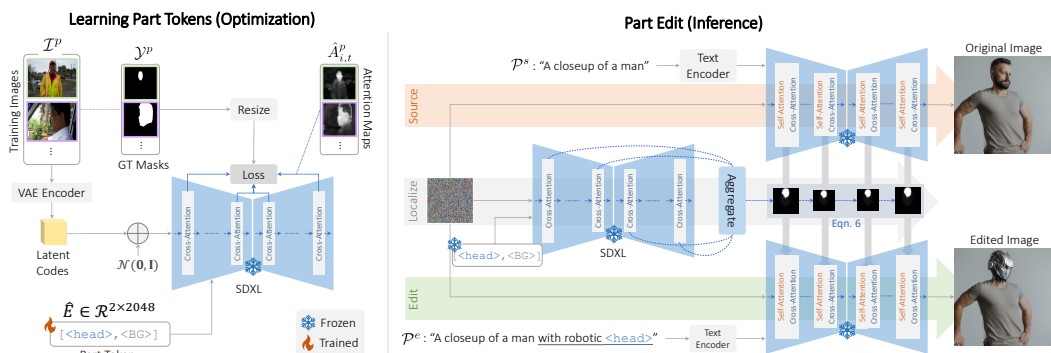

Figure 3: An overview of our proposed approach for fine-grained part editing. For an object part $p$, we collect a dataset of images $\mathcal{I}^p$ and their corresponding part annotation masks $\mathcal{Y}^p$. To optimize a textual token to localize this part, we initialize a random textual embedding $\hat{E}$ that initially generates random cross-attention maps. During optimization, we invert images in $\mathcal{I}^p$ and optimize the part token so that the cross-attention maps at different layers and timesteps match the part masks in $\mathcal{Y}^p$. After optimizing the token, it can be used during inference to produce a localization mask at each denoising step. These localization masks are used to perform feature bending between the source and the edit image trajectories. Note that we visualize three instances of SDXL (Podell et al., 2024) for illustration, but in practice, this is done with the same model in a batch of three.

new concepts while keeping the model's weights frozen. We leverage token optimization to perform fine-grained image editing of various object parts. We focus on optimizing tokens that can produce reliable non-binary blending masks *at each diffusion step* to localize the editing region. We supervise the optimization using either existing parts datasets such as PASCAL-Part (Donadello & Serafini, 2016), and PartImageNet (He et al., 2022) or a few user-annotated images.

## 3.1 Learning Part Tokens

Our training pipeline for object part tokens is illustrated in Figure 3. Given an object part $p$, we collect a set of images $\mathcal{I}^p = \{I^p_1, I^p_2, \ldots I^p_N\}$ and their corresponding segmentation masks of the respective parts $\mathcal{Y}^p = \{Y^p_1, Y^p_2, \ldots Y^p_N\}$. We start by encoding images in $\mathcal{I}^p$ into the latent space of a pre-trained conditional diffusion model using the VAE encoder and then add random Gaussian noise that corresponds to timestep $t_{start} \leq T$ in the diffusion process. Instead of conditioning the model on the embedding $E$ of a textual prompt as explained in Section 2, we initialize a random textual embedding $\hat{E} \in \mathrm{R}^{2 \times 2048}$ instead. This embedding $\hat{E}$ has two trainable tokens, where the first is optimized for the part of interest, and the second is for everything else in the image. Initially, this embedding will produce random cross-attention maps at different UNet blocks and timesteps.

To train a token for part $p$, we optimize the first token in $\hat{E}$ to produce cross-attention maps $\hat{A}^p_{i,t}$ that corresponds to the part segmentation masks in $\mathcal{Y}^p$ at different denoising steps $t$ and UNet blocks $i$. We employ the Binary Cross-Entropy (BCE) as a training loss for this purpose. For image $I^p_j \in \mathcal{I}^p$, a loss is computed as:

$$\mathcal{L}^p_j = \sum_t \sum_{i \in L} Y^p_j \log(\hat{A}^p_{i,t}) + (1 - Y^p_j) \log(1 - \hat{A}^p_{i,t}) \qquad (3)$$

where $t \in [t_{start}, t_{end}]$ are the diffusion timesteps that we include in the loss computation, and $L$ is the set of UNet layers. The loss is averaged over all pixels in $L^p_j$ and then over all images in $\mathcal{I}^p$. Note that the groundtruth segmentation masks $Y^p_j$ are resized to the respective size of the attention maps for loss computation. After optimizing the tokens, they are stored with the model as textual embeddings and are referred to as `<part-name>`. During denoising, these optimized tokens would produce a localization map for where the part is located in the image within the cross-attention modules.

Figure 4: The impact of the choice of which timesteps to include in the token optimization process. Intermediate timesteps achieve reasonable localization for both big and small parts.

## 3.2 CHOOSING TIMESTEPS AND UNET BLOCKS TO OPTIMIZE

Ideally, we would like to optimize over all timesteps and UNet layers. However, this is computationally and memory-expensive due to the large dimensionality of intermediate features in diffusion models, especially SDXL (Podell et al., 2024) that we employ. Therefore, we need to select a subset of layers and timesteps to achieve a good balance between localization accuracy and efficiency.

To determine the optimal values for $t_{start}, t_{end}$, we analyze the reconstructions of noise-free predictions $\hat{x}_0$ as per Equation (1) to observe the progression of image generation across different timesteps. Figure 4 shows that when optimizing over early timesteps $t_{start} = 50, t_{end} = 40$, the noise level is high, making it difficult to identify different parts. For intermediate timesteps, $t_{start} = 30, t_{end} = 20$, most of the structure of the image is present, and optimizing on these timesteps leads to good localization for big parts (*e.g.* the head) across most timesteps during inference. Moreover, the localization of small parts, such as the eyes, becomes more accurate the closer the denoising gets towards $t = 0$. Finally, optimizing on late timesteps, $t_{start} = 10, t_{end} = 0$, provides reasonable localization for big parts at intermediate timesteps but does not generalize well to early ones. Based on these observations, we choose to optimize on intermediate timesteps for localizing larger parts due to their consistent performance during inference time across all timesteps. For smaller parts, both intermediate and late timesteps offer satisfactory localization.

Regarding the selection of layers $L$ for optimization, we initially include all UNet blocks during training and subsequently evaluate each block's performance using the mean Intersection over Union (mIoU) metric. Our analysis reveals that the first eight blocks of the decoder are sufficient to achieve robust results, making them suitable for scenarios with limited computational resources. Further details are provided in the appendix.

After learning the part tokens, the diffusion model can now understand and localize parts through our optimized tokens. Next, we explain how we use them to perform fine-grained part edits. We start by describing our approach for the synthetic image setup where the diffusion trajectory is known; then, we explain how to perform real image editing.

## 3.3 PART EDITING

Given a source image $I^s$ that was generated with a source prompt $\mathcal{P}^s$, it is desired to edit this image according to the editing prompt $\mathcal{P}^e$ to produce the edited image $I^e$. To perform part edits, the editing prompt shall include one of the optimized part tokens that we refer to as <part-name>. As an example, a source prompt can be "A closeup of a man", and the editing prompt can be "A closeup of a man with a robotic <head>", where <head> is a part token. To apply the edit, we perform the denoising in three parallel paths (see Figure 3). The first path is the source path, which includes the trajectory of the original image that originates from a synthetic image or an inverted trajectory of a real image. The second path incorporates the part tokens that we optimized, and it provides the part localization masks. The final path is the edit path that is influenced by the two other paths to produce the final edited image. Since $\hat{E}$ has 2 tokens compared to 77 tokens in the source and the edit paths, we pad $\hat{E}$ with the background token to match the size of the other two embeddings. In the appendix, we provide more details on the choice of padding strategy.

For each timestep $t$ and layer $i$, we compute the cross-attention map $\hat{A}_t^i$ for the embedding $\hat{E}$ to obtain attention maps highlighting the part $p$. For timesteps $t < T$, the attention maps from the

previous step $t - 1$ are aggregated across all layers in $L$ to obtain a blending mask $M_t$ as follows:

$$M_t = \sum_{i \in L} \texttt{RESIZE}(\hat{A}_{t-1}^i) \,. \tag{4}$$

The mask $M_t$ is min-max normalized in the range $[0, 1]$. To perform a satisfactory edit, it is desired to edit only the part that is specified by the editing prompt, preserve the rest of the image, and seamlessly integrate the edit into the image. Therefore, we propose an adaptive thresholding strategy that fulfills this criterion given the aggregated mask $M_t$:

$$\mathcal{T}(X) = \begin{cases} 1 & \text{if } X \geq \omega \\ x & \text{if } k/2 \leq X < \omega \,, \\ 0, & X < k/2 \end{cases} \tag{5}$$
$$k = \texttt{OTSU}(X) \,.$$

where $\texttt{OTSU}$ is the OTSU thresholding (Otsu et al., 1975), $\omega$ is a tunable tolerance for the transition between the edited parts and the original object. We find that $\omega = 3k/2$ achieves the best visual quality, and we fix it for all experiments. This criterion ensures suppressing the background noise and a smooth transition between the edited part and the rest of the object. Finally, we employ $M_t$ to blend the features between the source and the editing paths as:

$$\hat{F}_{i,t}^e = \mathcal{T}(M_t) \, \hat{F}_{i,t}^e + (1 - \mathcal{T}(M_t)) \, \hat{F}_{i,t}^s \tag{6}$$

where $\hat{F}_{i,t}^s$ and $\hat{F}_{i,t}^e$ are the image features after attention layers for the source and the edited image, respectively. We apply this blending for timesteps in the range $[1, t_e]$ where $t_e \leq T$ and the choice of $t_e$ controls the locality of the edit. More specifically, a higher $t_e$ indicates higher preservation of the unedited regions, while a lower $t_e$ gives the model some freedom to add relevant edits in the unedited regions. We provide more details in Section 4.4.

## 3.4 REAL IMAGE EDITING

Our approach can also perform part edits on real images by incorporating a real image inversion method, *e.g.*, Ledits++ (Brack et al., 2024) or EF-DDPM (Huberman-Spiegelglas et al., 2024). In this setting, the role of the inversion method is to estimate the diffusion trajectory $x_0 \ldots x_T$ for a given real image. This estimated trajectory is then used as the source path in Figure 3. To obtain the source prompt $\mathcal{P}^s$ of the real images, we use the image captioning approach, BLIP2 (Li et al., 2023), which is commonly used for this purpose. Finally, the edit is applied where the localization and editing paths are similar to the synthetic setting.

## 3.5 DISCUSSION

Our approach is a text-based editing approach eliminating the need for the user-provided masks to perform fine-grained edits. Despite the fact that the token optimization process requires annotated masks for training, it can already produce satisfactory localization based on a single annotated image (see the Appendix). Moreover, the localization masks produced by the tokens are non-binary, leading to a seamless blending of edits, unlike user-provided masks, which are typically binary.

Another aspect is that our approach can be used in the context of image generation in case of complicated concepts that are difficult for diffusion models to comprehend from regular prompts. Examples of this scenario were shown in Figure 2, where a direct generation process failed to generate the requested concepts. In that case, and with the help of the optimized part tokens, the user can specify different attributes for different parts of the object in the image.

## 4 EXPERIMENTS

In this section, we provide an evaluation of our proposed approach, a comparison against existing text-based editing approaches, and an ablation study. To facilitate these aspects, we create a manually annotated dataset of several object parts. For a comprehensive evaluation, we compare against both *synthetic* and *real* image editing approaches.

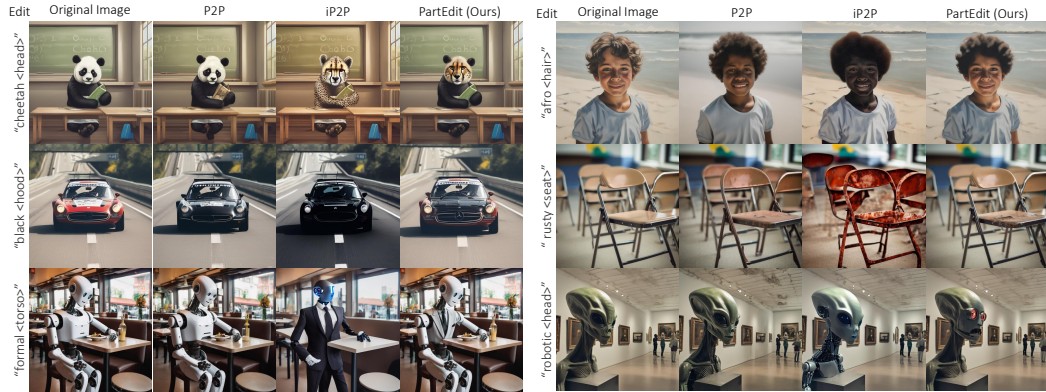

Figure 5: A qualitative comparison on synthetic images from the PartEdit dataset.

## 4.1 EVALUATION SETUP

**Synthetic Image Editing** We base our method on a pre-trained SDXL (Podell et al., 2024) [1]. For sampling, we employ the DDIM sampler (Song et al., 2020) with $T = 50$ denoising steps and default scheduling parameters as (Hertz et al., 2022). For token optimization, we train on 10-20 images per token for 2000 optimization steps. We set $t_e = T = 50$ for complete preservation of the unedited regions. We compare against two popular text-based editing approaches: Prompt-to-Prompt (P2P) (Hertz et al., 2022) and Instruct-Pix2Pix (iP2P) (Brooks et al., 2023). We use the SDXL implementation of P2P[2], and the Diffusers implementation of iP2P[3].

**Real Image Editing** We use Ledits++ (Brack et al., 2024) inversion method to invert the images where the source prompt $\mathcal{P}^s$ is produced by BLIP2 (Li et al., 2023) as described in Section 3.4. We compare against Ledits++ and EF-DDPM (Huberman-Spiegelglas et al., 2024)

**PartEdit Dataset** We create a synthetic and real dataset of 7 different object parts: `<humanoid-head>`, `<humanoid-torso>`, `<human-hair>` `<animal-head>`, `<car-body>`, `<car-hood>`, and `<chair-seat>`. For the synthetic dataset, we generate random source prompts of the objects of interest at random locations and select random edits from pre-defined lists. We generate a total of 60 synthetic images and manually annotate the part of interest. For the real dataset, we collect 20 images from the internet and manually annotate and assign editing prompts to them. We denote the synthetic and real datasets as *PartEdit-Synth* and *PartEdit-Real*, respectively. More details are provided in the Appendix.

**Evaluation Metrics** To evaluate the edits, we want to verify that: (1) The edit has been applied at the correct location. (2) The unedited regions and the background have been preserved. Therefore, We evaluate the following metrics for the foreground (the edit) and the background:

1. $\alpha\text{Clip}_{FG}$: the CLIP similarity between the editing prompt and the edited image region.
2. $\alpha\text{Clip}_{BG}$: the CLIP similarity between the unedited region of the source and the edited image. We also compute *PSNR* and structural similarity *(SSIM)* for the same region.

where $\alpha\text{Clip}$ is the masked CLIP from (Sun et al., 2024).

## 4.2 QUALITATIVE RESULTS

**Synthetic Image Editing** Figure 5 shows a qualitative comparison. Our approach excels at performing challenging edits seamlessly while perfectly preserving the background. P2P fails in most cases to perform the edit as it cannot localize the editing part. iP2P completely ignores the specified part and applies the edit to the whole object in some cases and produces distorted images in other cases.

---

[1] Our code and dataset will be made publicly available.

[2] https://github.com/RoyiRa/prompt-to-prompt-with-sdxl

[3] https://huggingface.co/docs/diffusers/en/api/pipelines/pix2pix

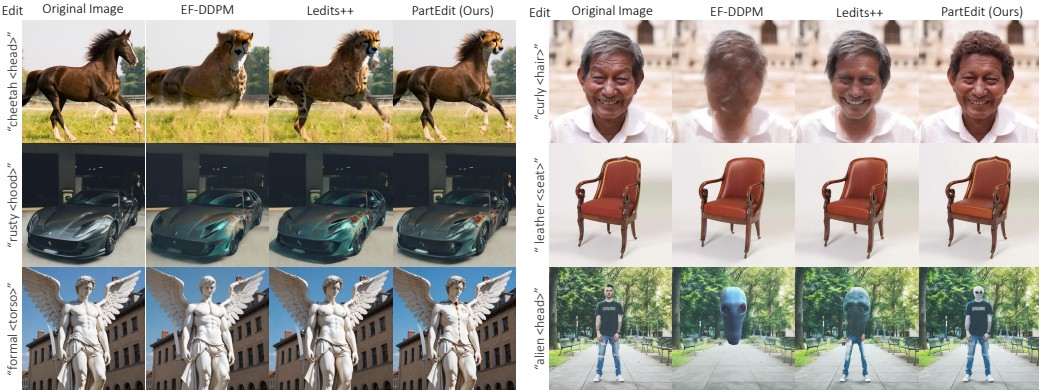

Figure 6: A qualitative comparison against EF-DDPM (Huberman-Spiegelglas et al., 2024) and Ledits++ (Brack et al., 2024) on real image editing.

Remarkably, our approach succeeds in eliminating racial bias in SDXL by decoupling the hair region from the skin color, as shown in the top-right example.

**Real Image Editing** Figure 6 shows a qualitative comparison of real image editing. PartEdit produces the most seamless and localized edits, whereas other approaches fail to localize the correct editing region accurately. Additionally, we provide a qualitative comparison against InfEdit (Xu et al., 2024) and PnPInversion (Ju et al., 2024) in the supplementary.

### 4.3 QUANTITATIVE RESULTS

**Synthetic Image Editing** Table 1 top summarizes the quantitative metrics on *PartEdit-Synth* dataset. Our approach performs the best on $\alpha\text{Clip}_{\text{avg}}$ and all metrics for the unedited regions. iP2P scores the best in terms $\alpha\text{Clip}_{\text{FG}}$ as it tends to change the whole image to match the editing prompt, ignoring the structure and the style of the original image (see Figure 5). We also performed two user studies comparing our approach against P2P and iP2P on Amazon Mechanical Turk. Our approach is preferred by the users 88.6 % and 77.0% of the time compared to P2P and iP2P, respectively. The reported results for the user studies are based on 360 responses per study.

**Mask-Based Editing** To demonstrate the effectiveness of our text-based editing approach, we compare it against two mask-based editing approaches, SDXL inpainting and SDXL Latent Blending (Avrahami et al., 2023b), where the user provides the editing mask. We use the groundtruth annotations to perform the edits for these two approaches, while our approach relies on the optimized tokens. Table 1 shows that our approach is preferred by 75% and 66% of users over the mask-based approaches. This reveals that our editing approach produces visually more appealing edits even compared to the mask-based approaches, where the mask is provided.

**Real Image Editing** Table 1 shows that PartEdit outperforms Ledits++ and EF-DDPM on all metrics and is significantly favored by users in the user study. This demonstrates the efficacy of our approach in performing fine-grained edits. Additionally, we compare against InfEdit (Xu et al., 2024) and PnPInversion (Ju et al., 2024).

### 4.4 ABLATION STUDY

**Impact of $t_e$** The choice of the number of timesteps to perform feature blending using Equation (6) controls the locality of the edit, where the blending always starts at timestep one and ends at $t_e$. Figure 7 shows two different scenarios for how to choose $t_e$ based on the user desire. In the first row, a low $t_e$ causes the horse's legs and tail to change to dragon ones, but a higher $t_e$ would change only the head and preserve everything else. The second row has another example: a lower $t_e$ discards the girl's hair, and a higher $t_e$ preserves it. Consequently, $t_e$ gives the user more control over the locality of the performed edits.

| Method | Edited Region | Unedited Region | | | $\alpha\text{Clip}_{avg}\uparrow$ | Ours Pref. (%)$\uparrow$ |
|---|---|---|---|---|---|---|
| | $\alpha\text{Clip}_{FG}\uparrow$ | $\alpha\text{Clip}_{BG}\uparrow$ | PSNR $\uparrow$ | SSIM $\uparrow$ | | |
| *Synthetic Image Editing* | | | | | | |
| P2P | 63.60 | 20.46 | 23.7 | 0.87 | 42.03 | 88.6 |
| iP2P | 89.31 | 5.55 | 18.0 | 0.74 | 47.43 | 77.0 |
| PartEdit (Ours) | **91.74** | **38.38** | **31.5** | **0.98** | **65.06** | - |
| *Mask-Based Editing* | | | | | | |
| SDXL Inpainting | 79.72 | 46.76 | 34.2 | 0.96 | 63.24 | 75.0 |
| Latent Blending | 90.09 | 46.45 | 37.5 | 0.98 | 68.27 | 66.1 |
| *Real Image Editing* | | | | | | |
| Ledits++ | 66.57 | 12.95 | 20.9 | 0.69 | 39.76 | 80.77 |
| EF-DDPM | 73.57 | 15.09 | 22.9 | 0.72 | 44.33 | 90.38 |
| InfEdit | 67.67 | 21.71 | 22.5 | 0.70 | 44.69 | 77.88 |
| PnPInversion | 71.71 | 23.59 | 22.6 | 0.73 | 47.65 | 79.81 |
| Ours w/ Ledits++ | **85.59** | **28.29** | **26.3** | **0.76** | **56.94** | - |

Table 1: A quantitative comparison on the task of parts editing. Ours Pref. indicates the percentage where our approach was favored by users in the user study.

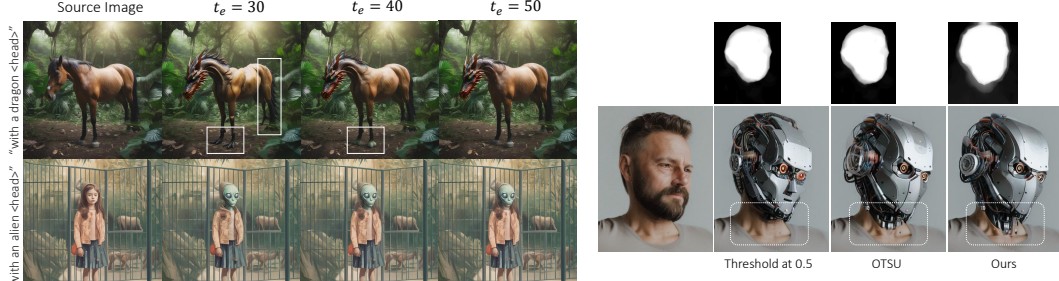

Figure 7: The impact of changing the number of denoising steps $t_e$ to perform feature blending.

Figure 8: Comparison of an edit under different mask binarization strategies.

**Impact of Binarization** To show the efficacy of our proposed thresholding strategy in Equation (5), we show an edit under different binarization strategies in Figure 8. Standard binarization, where the blending mask is thresholded at 0.5, leads to the least seamless edit as if a robotic mask was placed on the man's head. OTSU integrates some robotic elements on the neck, but they are not smoothly blended. Finally, our thresholding strategy produces the best edit, where the robotic parts are seamlessly integrated into the neck.

## 5 RELATED WORK

### 5.1 DIFFUSION-BASED IMAGE EDITING

In general, the image semantics are encoded within the cross-attention layers, which specify where each word in the text prompt is located in the image (Hertz et al., 2022; Tang et al., 2023). In addition, the style and the appearance of the image are encoded through the self-attention layers (Tumanyan et al., 2023). Diffusion-based editing approaches exploit these facts to perform different types of semantic or stylistic edits. On the semantic level, several approaches attempt to change the contents of an image according to a *user-provided prompt* (Hertz et al., 2022; Brooks et al., 2023; Lin et al., 2023; Kawar et al., 2023; Parmar et al., 2023). These changes include swapping an object or altering its surroundings by manipulating the cross-attention maps either through token replacement or attention-map tuning. For stylistic edits, several approaches (Cao et al., 2023; Tumanyan et al., 2023; Parmar et al., 2023; Hertz et al., 2023) modify the self-attention maps to apply a specific

style to the image while preserving the semantics. This style is either provided by the user in the form of text or a reference image. It is worth mentioning that an orthogonal research direction investigates image inversion to enable real image editing (Brack et al., 2024; Huberman-Spiegelglas et al., 2024; Brooks et al., 2023). Another line of research focuses on memory-based guidance for moving objects (dragging) (Mou et al., 2023). Methods that focus on obtaining a fine-grained mask or editing segments of the image (Liu et al., 2024; Yu et al., 2024; Singh et al., 2023; Wei et al., 2024; Kirillov et al., 2023). For a comprehensive review of editing techniques and applications, we refer readers to (Huang et al., 2024). Existing text-based editing approaches struggle to apply semantic or stylistic edits to fine-grained object parts, as demonstrated in Section 4. Our approach, PartEdit, is the first step toward enabling fine-grained editing, which will enhance user controllability and experience, and can potentially be integrated into existing editing pipelines.

## 5.2 Token Optimization

To expand the capabilities of pre-trained diffusion models or to address some of their limitations, they either need to be re-trained or finetuned. However, these approaches are computationally expensive due to the large scale of these models and the training datasets. An attractive alternative is token optimization, where the pre-trained model is kept frozen, and special textual tokens are optimized instead. Those tokens are then used alongside the input prompt to the model to perform a specific task through explicit supervision of cross-attention maps. This has been proven successful in several tasks. (Valevski et al., 2023; Gal et al., 2022; Avrahami et al., 2023a; Safaee et al., 2024) employed token optimization and LoRA finetuning to extract or learn new concepts that are used for image generation. (Hedlin et al., 2023) learned tokens to detect the most prominent points in images. (Marcos-Manchón et al., 2024; Khani et al., 2024) optimized tokens to perform semantic segmentation. We explored the use of token optimization to learn tokens for object parts customized with a focus on image editing. For this purpose, we adopt SDXL (Podell et al., 2024) as a base model to obtain the best visual editing quality in contrast to existing approaches that leverage SD 1.5 or 2.1. Our token optimization training is also tailored to obtaining smooth cross-attention maps across all timesteps, unlike (Marcos-Manchón et al., 2024; Khani et al., 2024) that aggregate all attention maps and apply post-processing to obtain binary segmentation masks.

## 6 Concluding Remarks

**Limitations** Our approach is limited by the generation capability of the underlying diffusion model. For instance, it can not perform unreasonable edits that combine irrelevant parts, *e.g.*, editing a human head to become a car wheel. Moreover, it can not change the style of the edited part to a different style other than the style of the original image. This limitation stems from the internal design of SDXL that encodes the style in self-attention and prevents mixing two different styles. We provide examples of these scenarios in the Appendix.

**Ethical Impact** Our approach can help alleviate the internal racial bias of some edits, such as correlating "Afro" hair with Africans, as we demonstrated in Figure 5. On the other hand, our approach can produce images that might be deemed inappropriate for some users by mixing parts of different animals or humans. Moreover, the remarkable seamless edits performed by our approach can potentially be used for generating fake images, misinformation, or changing the identity.

**Future Work** Our approach can be extended to perform 3D edits similar to Instruct-Nerf2Nerf Haque et al. (2023). An LLM can also be incorporated to parse complicated editing prompts to the correct part token. For example, mapping "A man with a muscular upper body" to the token "`<torso>`".

## 7 Conclusion

We introduced the first text-based editing approach for object parts based on a pre-trained diffusion model. Our approach can perform appealing edits that possess high quality and are seamlessly integrated with the parent object. Moreover, it can create concepts that the standard diffusion models and editing approaches are incapable of generating. This helps to unleash the creativity of creators, and we hope that our approach will establish a new line of research for fine-grained editing approaches.

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

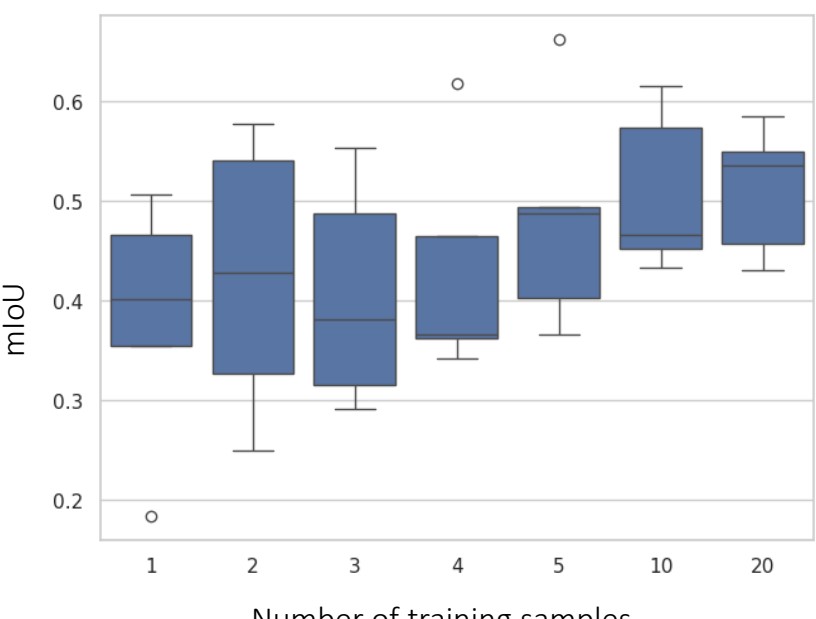

Figure 9: The impact of the number of training samples on mIoU.

## A   APPENDIX

## B   ADDITIONAL QUALITATIVE RESULTS

We provide more qualitative results in the attached supplementary material.

## C   NUMBER OF TRAINING SAMPLES

To study the impact of the number of training samples used for training each individual token, we train on varying numbers of samples from the quadruped-head training set of the PartImageNet dataset. We then compute the mean intersection-over-union (mIoU) on 50 randomly sampled validation samples. Figure 9 shows the mIoU over 5 runs that are optimized for 2000 steps. The figure shows that 10-20 training samples achieve a similar mIoU. The mIoU might improve further with more samples, but we observed that this small number of samples is sufficient to attain a good localization of different parts. This is a clear advantage of our approach compared to training a part segmentation model that requires a large number of samples.

## D   CHOICE OF $\Omega$ FOR OUR ADAPTIVE THRESHOLDING

To understand how the choice of $\Omega$ in Equation 5 affects the editing, we provide an example for an edit with different values of $\Omega$ in Figure 10. Lower values of $\omega$ make the editing regions dominated by the editing prompt and put less emphasis on blending with the main object. This is demonstrated by the Joker's head, which does not follow the original head pose. By increasing the value of $\Omega$, the blending starts to improve, and the edit becomes more harmonious with the original object. At $\Omega = 3k/2$, the best trade-off between performing the edit and blending with the original object is achieved, and we find it to be optimal for most edits. Increasing $\Omega$ further can make the original object dominate over the edit, *e.g.* the hair of the man remains unedited.

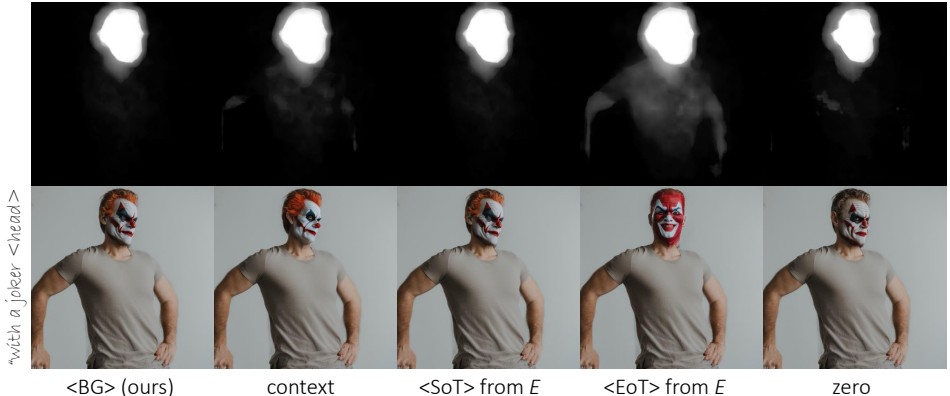

Figure 10: Applying the edit "with a Joker <head>" with different choices of hyperparameter $\omega$.

Figure 11: Influence of different token padding strategies during inference on the cross-attention maps.

## E CHOICE OF TOKEN PADDING STRATEGIES

During token optimization, we train a custom embedding $\hat{E} \in \mathrm{R}^{2 \times 2048}$. However, during inference, the dimensionality of this embedding needs to match the standard SDXL embedding $E \in \mathrm{R}^{77 \times 2048}$. Therefore, our embedding needs to be padded with some tokens to match that of SDXL. Possible choices are: padding with [2-77] tokens from $E$ (context), zero padding, <BG> token, <SoT> from $E$, or <EoT> token from $E$. We display the effect of these different strategies on the average extracted attention map in Figure 11 The figure shows that padding with the <BG> or <SoT> from $E$ attains the cleanest attention maps, leading to the best edits in terms of blending with the main object.

## F CHOICE OF UNET LAYERS FOR TOKEN OPTIMIZATION

To decide which UNet layers to include in token optimization ($L$ in Equation 3), we compute the mIoU for every individual layer based on their cross-attention maps. The results are shown in Figure 12. We can observe that the first 8 layers of the Decoder with indices [24,32] achieve the best mIoU, indicating that they are semantically rich. Those 8 layers can be used to optimize the tokens rather than all layers in case of limited computational resources.

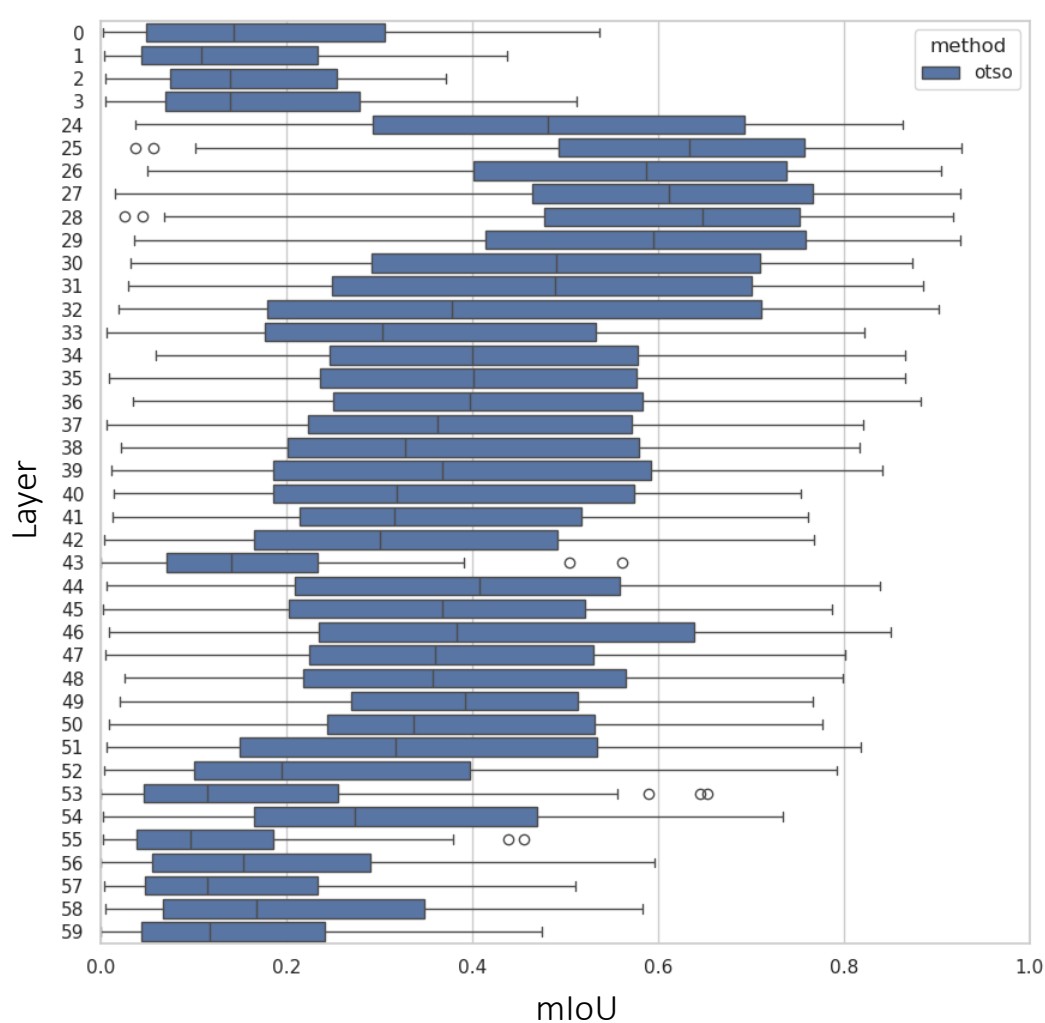

Figure 12: Analysis of how each layer of the UNet performs in terms of mioU. The first eight collected layers of the decoder achieve the best results. Note that we apply OTSU for binarization.

## G  DIFFERENT EDITS PER IMAGE

To show that our method performs consistently well with different edits, we provide some qualitative examples. In Figure 13, we apply multiple identity edits to the input image. This figure showcases the powerful versatility of our approach. Two noteworthy edits are "young" and "old", where the identity of the man is preserved and aged according to the edit. Moreover, our approach successfully changes the identity of multiple celebrities of different ethnicities seamlessly at an exceptional quality.

We provide other examples in Figure 14 for real images, where different edits are applied. The figure shows that our approach performs consistently well and can apply edits of a different nature to the same image.

## H  FAILURE CASES

Figure 15 shows two examples of failure cases. The first example shows that the edits performed by our approach are restricted to the style of the original image. More specifically, it can not change the style from "real" to a "drawing". This limitation stems from the internal design of SDXL that encodes the style in self-attention and prevents mixing two different styles. The second example

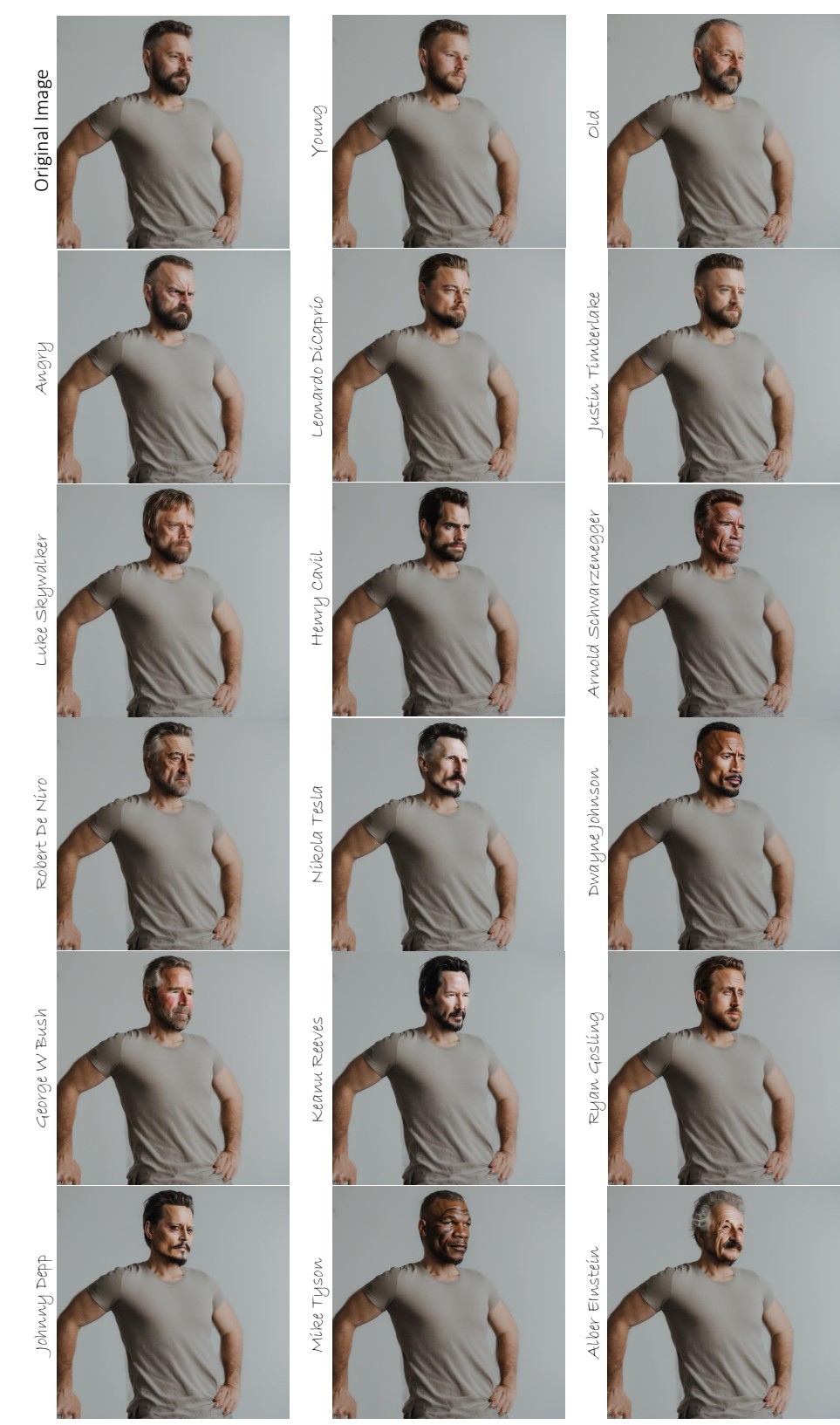

Figure 13: Applying different identity edits to the same image.

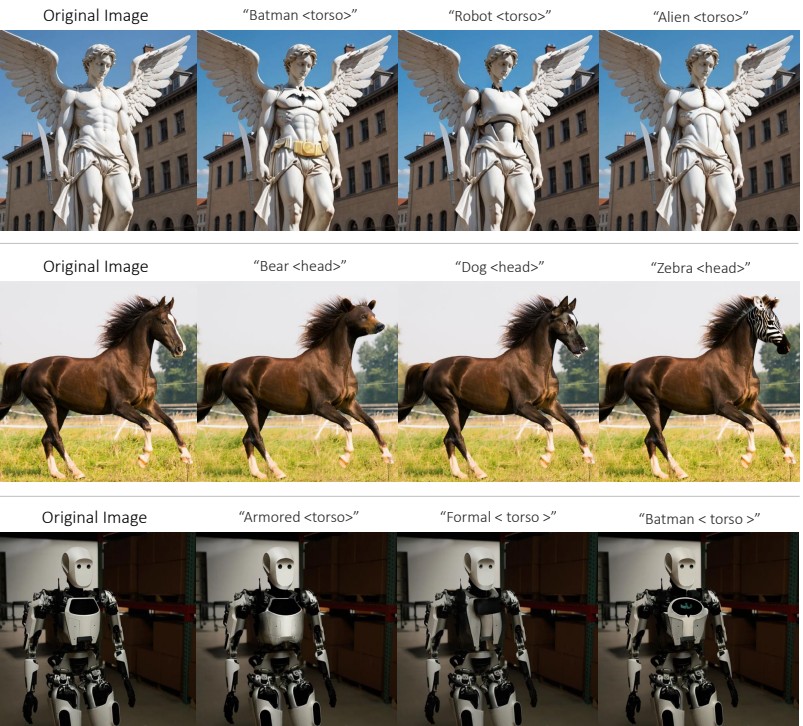

Figure 14: Different edits per image on real image editing.

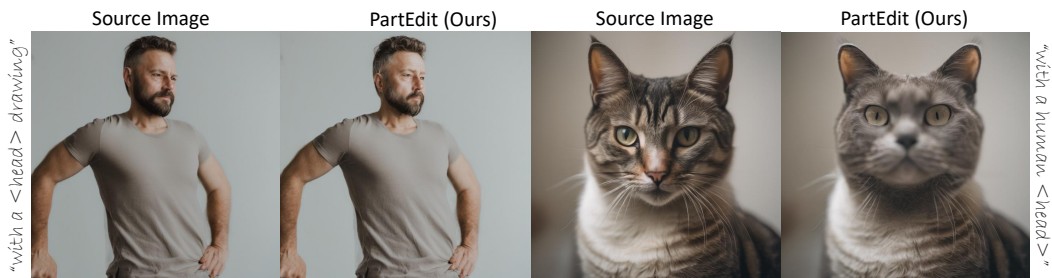

Figure 15: Failure cases. Our approach can not perform an edit in a different style (left) or unreasonable edits (right).

shows that our approach can not perform unreasonable edits, such as replacing a cat's head with a human head. This limitation arises from the incapability of SDXL to generate these concepts, but we see a promising direction of disentanglement of existing concepts.

## I    USER STUDY DETAILS

We conducted two user studies against P2P and iP2P independently using the 2AFC technique with a random order of methods. We used Amazon Web Turk, with the minimal rank of "masters," and received 360 responses per study. The users were provided with the following instructions: "Y"ou are given an original image on the left, edited using the A and B methods. Please select the method that changes ONLY the part specified by PART and keeps the rest of the image unchanged. For example, if the original image has a 'Cow", PART is "Head", and EDIT is "Dragon", choose the method that changes the cow head to dragon head, and keeps the rest of the cow's body as it is. A screenshot of the user study layout is shown in Figure 16.

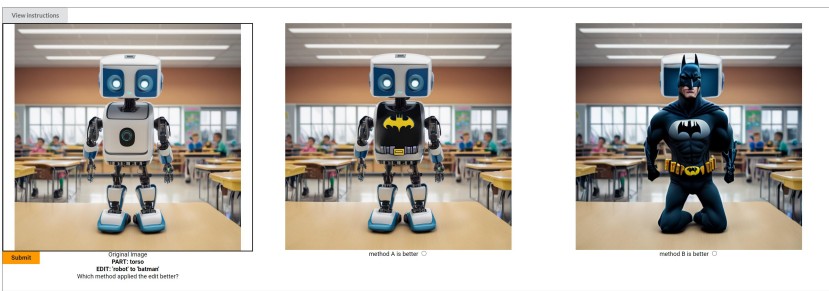

Figure 16: Visualization of the user study layout that we conducted.

## J  GENERATING PROMPTS AND EDITS FOR EVALUATION

We provide an example of how we generate prompts and edits for evaluation of <animal-head> in Figure 17. We follow the same strategy for all other parts.

## K  IMPACT OF CHOICE OF IMAGES FOR TRAINING PART TOKENS

We further validate the impact of the choice of images during training. We perform a cross-validation experiment using SD2 with the <Quadruped head> (PartImageNet). Specifically, we utilize 100 images in a 5-fold cross-validation setup, achieving a mean IoU of $71.704$ with a standard deviation of $4.372$. This highlights the stability and generalization of the model's semantic part segmentation with optimized tokens across different training subsets.

## L  HYPERPARAMETERS

We provide hyperparameters used for training the tokens and hyper parameters used during editing.

For the training process, we deploy BCE loss of initial learning rate of $30.0$, with a StepLR scheduler of $80$ steps size and gamma of $0.7$. For the diffusion, we use a strength of $0.25$ and a guidance scale of $7.5$. During aggregation for loss computation, masks are resized to $512$ for SD2.1 and $1024$ for SDXL model variants. We use 1000 or 2000 epochs during training.

During inference, as discussed in Figure 10, we use $\omega$ of $1.5$. For the guidance scale, larger values tend to increase adherence to $\alpha CLIP$ but at the cost of PSNR and SSIM. We investigated values between 3.5 and 20 for the guidance scale and used 12.5 as a balance between the two for Ours + edits real setting. For the synthetic setting, a guidance scale of 7.5 was used. During inference, we start editing at the first step, as we utilize prior time step mask information.

## M  ADDITIONAL QUALITATIVE COMPARISONS

We provide additional qualitative comparisons with InfEdit and PnPInversion for the same images as in the main paper (in addition to quantitative results in the main table), and we can observe that our approach outperforms them. Nonetheless, InfEdit does showcase potential with the Alien head example Figure 18. For both methods, we use the default parameter values provided in their demo/code, with default blending words between the object and their edited instruction. (E.g., source prompt is "A statue of an angel with wings on the ground," target prompt is "A statue with the uniform torso of an angel with wings on the ground," and blend between "statue" and "statue with uniform torso").

## N  EDITING MULTIPLE REGIONS AT ONCE

Our approach can be easily adapted to edit multiple parts simultaneously at inference time without retraining the tokens. To achieve this, the part tokens are loaded and fed through the network to

```json
{
    "template": "A <OBJECT> at a <LOCATIONS>",
    "template2": "A <EDIT> at a <LOCATIONS>",
    "objects": [
        "cat",
        "dog",
        "monkey",
        "bird",
        "bear",
        "panda",
        "horse",
        "cow",
    ],
    "locations": [
        "forest",
        "city",
        "farm",
        "park",
        "desert",
        "jungle",
        "house",
        "restaurant",
        "library",
        "museum",
        "theater",
        "stadium",
        "pool"
    ],
    "edits":{
        "_animal":[
            "panda",
            "cat",
            "dog",
            "lion",
            "tiger",
            "cheetah",
            "dragon",
            "unicorn",
        ],
    }
}
```

Figure 17: Example of configuration for random generation of the edits.

produce cross-attention maps at different layers of the UNet. We accumulate these maps across layers and timesteps as described in Section 3.3, but the main difference is that we normalize the attention maps for different parts jointly at each layer. We provide several examples in Figure 19 and visualizations of the combined attention maps.

## O  ADDITIONAL ATTENTION MAP LOCALIZATION VISUALIZED

Here, we provide additional visualizations of images, their annotated ground truth, raw normalized token attention maps, binarized attention maps using a $0.5$ threshold, binarized attention maps using the Otsu threshold, and our approach using Otsu threshold ($\omega = 1.5$). One can observe that our approach can be thought of as a relaxation of binary thresholding but a stricter version of Otsu thresholding.

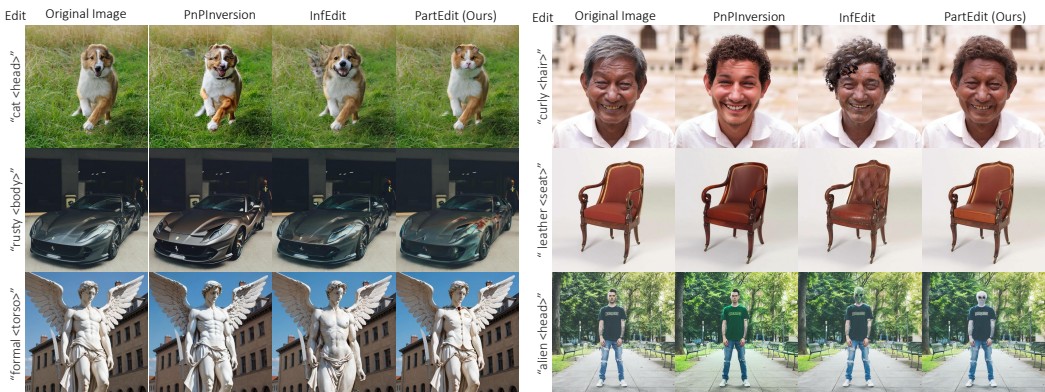

Figure 18: InfEdit (Xu et al., 2024) and PnPInversion (Ju et al., 2024) on real image setting.

## P   USE OF EXISTING SEGMENTATION MODELS LIKE SAM

Segment Anything Kirillov et al. (2023), as one of the foundational models in segmentation using conditioned inputs such as points, still struggles to segment parts that do not have a harsh border (commonly torso and head) while it has no problems with classes such as car hood. We can observe such failure cases in Figure 21.

## Q   USER INTERFACE

We provide an illustration of our user interface in Figure 22. The user specifies the editing prompt in the form "with <edit> <part-name>", and the corresponding token is loaded to apply the desired edit. The user also has the option to tune the $t_e$ parameter (Editing Steps) to control the locality of the edit. We also visualize the aggregated editing mask to help the user understand the results.

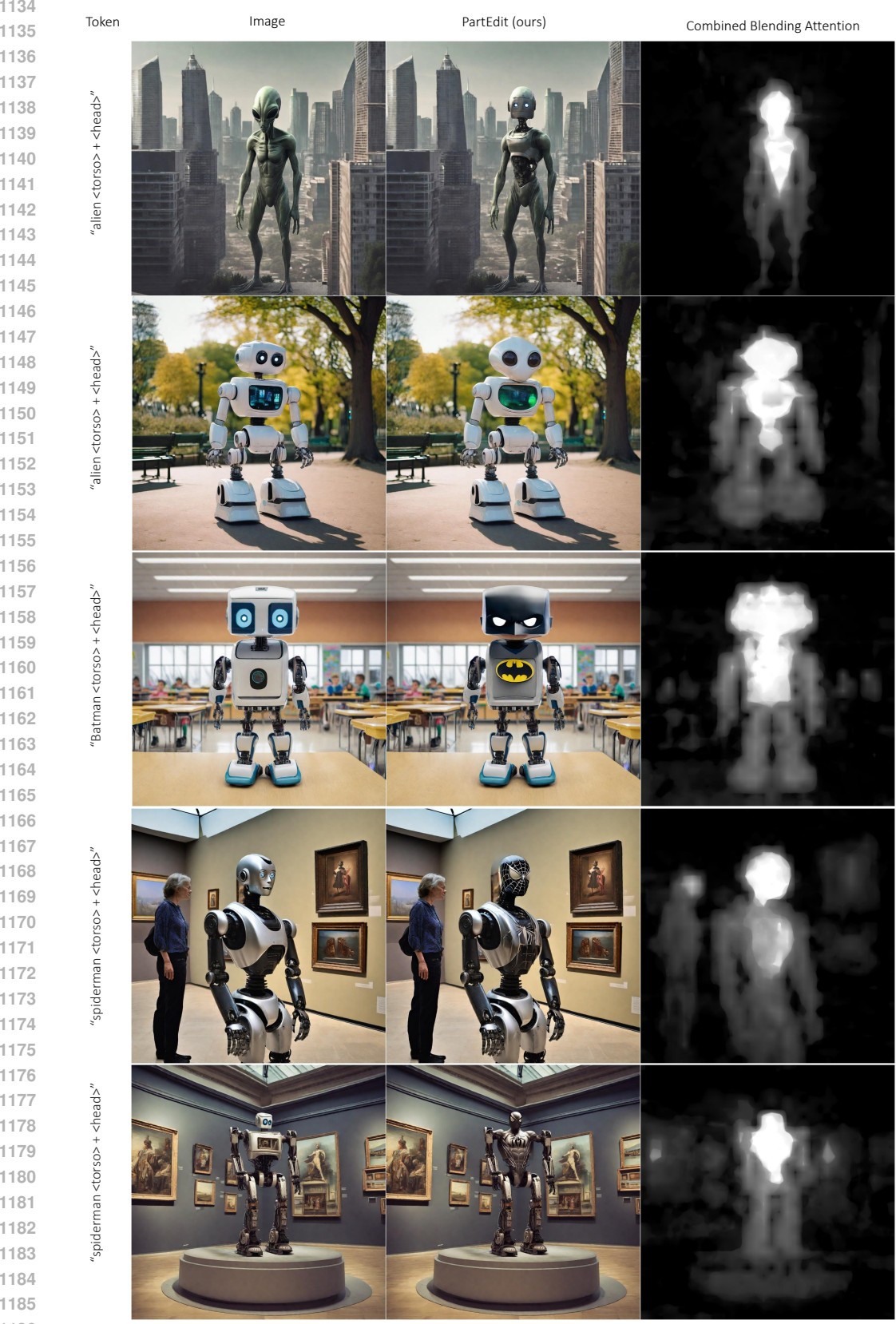

Figure 19: Visualization of editing 2 parts at the same time.

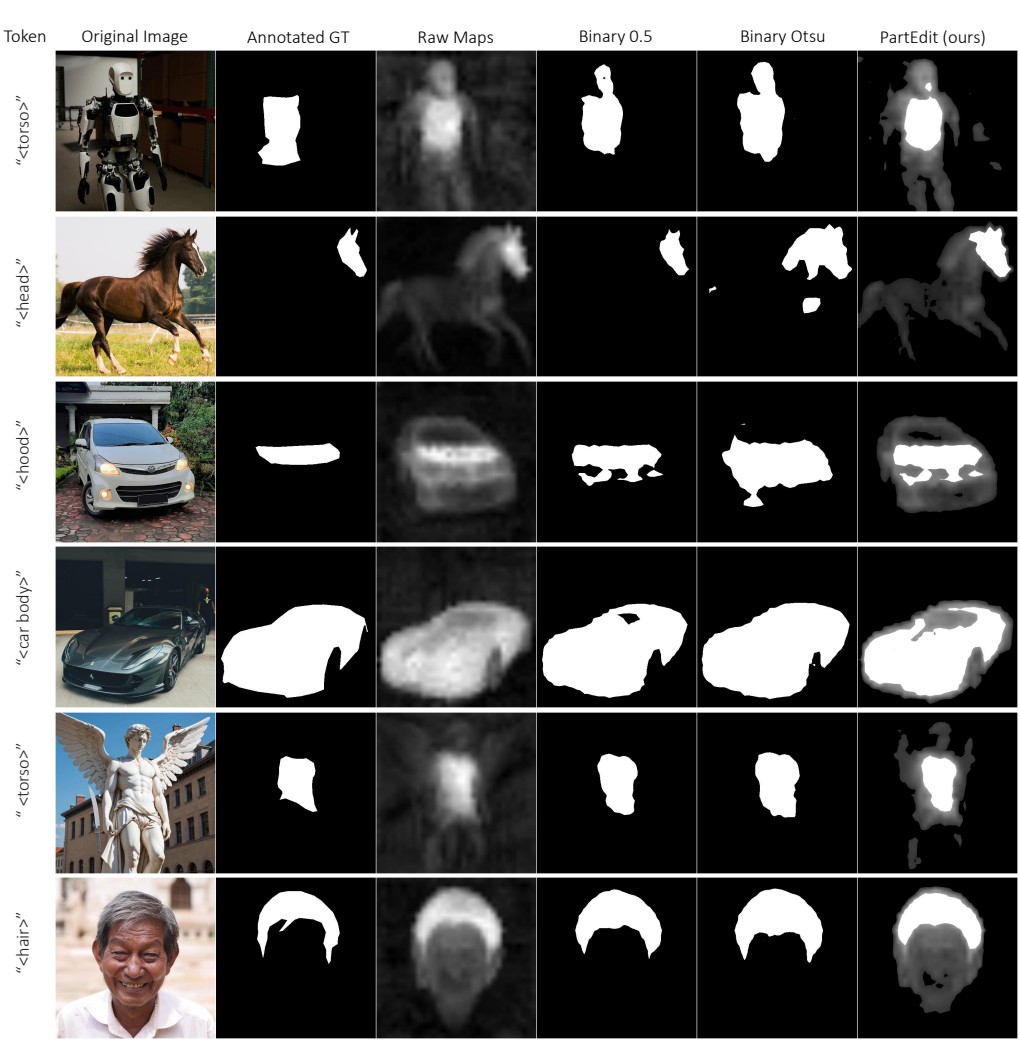

Figure 20: Additional visualization of obtained attention maps across all time steps of the qualitative results under the real setting.

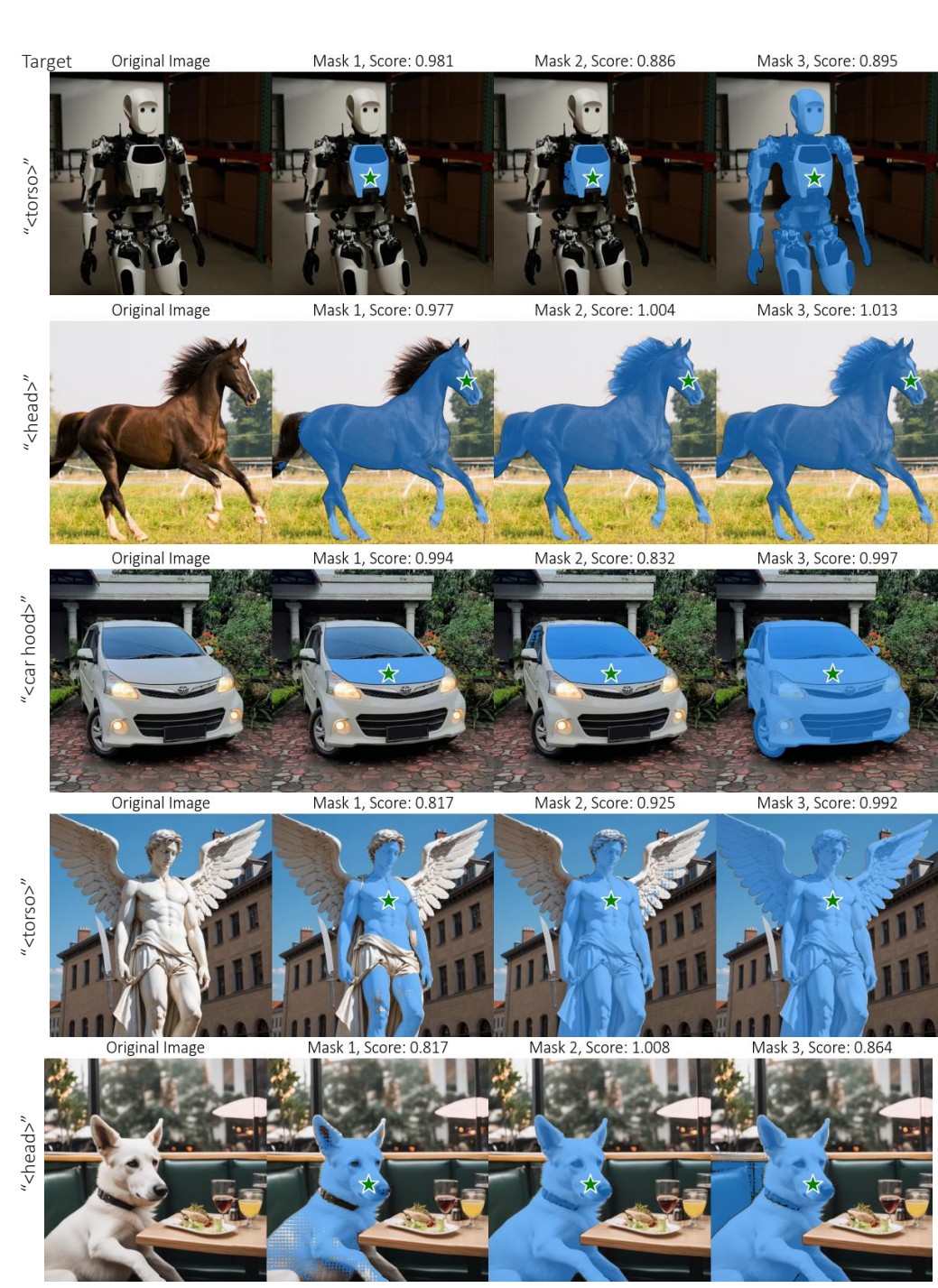

Figure 21: Visualization of masks obtained from Segment Anything (huge) model across the 3 heads for the green provided point. The target indicates what we wanted to segment by the green point.

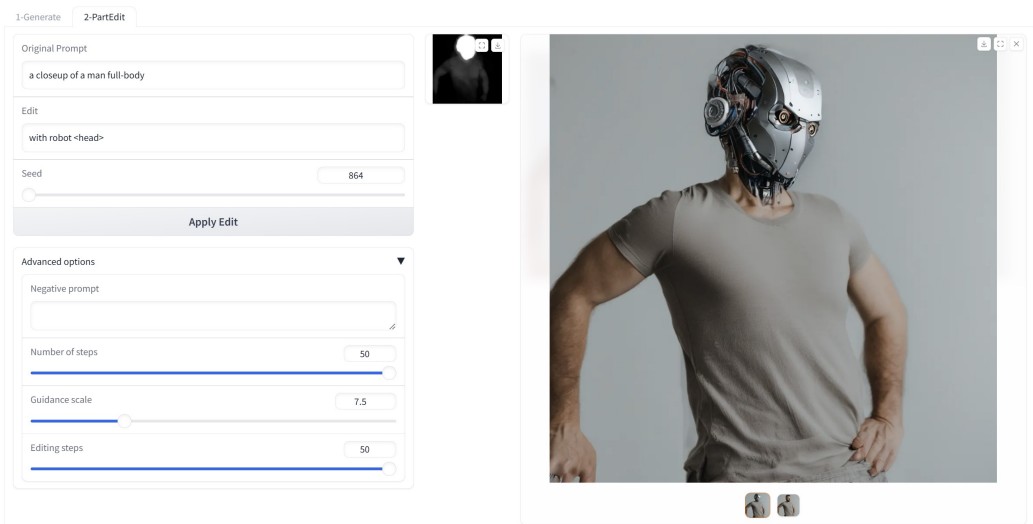

Figure 22: An illustration of our user interface.

