# OpenReview forum: "PartEdit: Fine-Grained Image Editing using Pre-Trained Diffusion Models"
_ICLR.cc/2025/Conference — Submitted to ICLR 2025_

### Official Review · Reviewer_kfJM · 2024-10-23

**Soundness:** 3
**Presentation:** 3
**Contribution:** 2
**Rating:** 3
**Confidence:** 4

**Summary:**

The paper proposes a method for part editing in images. The paper shows that current state-of-the-art fails when asked to change only particular parts of images (e.g. 'hood' of a car). The paper proposes to perform part token learning, and then uses the attention maps of the learned part tokens for accurate part-based editing of images. Results show improved results over several baseline methods.

**Strengths:**

- the objective of accurate part editing with text prompts is relevant for many users.
- the method is simple and results show that the method successfully addresses the problem.
- overall writing and presentation is good.

**Weaknesses:**

- I find the main scientific/technologic contribution of the paper insufficient for ICLR conference, nor does the paper provided new insights in the functioning of DM for editing. I agree part-based editing is relevant. Given existing methods, part-based editing can be addressed by solving part detection or just user provided masks (for example based on segment anything). The proposed method of learning part tokens makes sense. The computed attention maps reduce the problem to a text-based inpainting problem.
- the method is only applied to a very limited set of parts (7). Are these stored in 7 different models or jointly held within a single network ? Could this scale to many more parts ? Some analysis of the quality as a function of number of parts (if contained in a single model) would be interesting.
- the method needs to learn new prompts for every new part users might want to change. The method dependents on existing part datasets for these parts, else they need to be created. Do the authors see any other solutions, using other existing models to preven t annotation?

minor
- are all weights tuned, or do you use LoRA for layer optimization of the L layers
- figure 3 could be improved, it is hard to read in print

**Questions:**

See weaknesses.

I think is not of ICLR quality (the scientific contribution is too small) and could be published in a more applied venue (e.g. WACV) or dedicated workshop.

---

> ### Author Response · Authors · 2024-11-24
>
> W1 - Text-based editing is the most convenient and user-friendly form of editing as it does not require any skill or low-level interaction from the user.
> Our paper introduces the first fine-grained text-based editing approach that is highly beneficial for image editing research as noted by the reviewers: SEmy: “This paper focus on an interesting question, which as great significance to downstreaming research and tasks.” , 8GnP: “This paper addresses a critical problem in image editing…”, and rJwB: “The paper presents a flexible method for text-based image editing focused on object parts, which is a novel contribution to the field of image processing and editing.”
> For mask-based editing, Segment-Anything (SAM) model usually struggles with segmenting parts as the model does not know if the user is interested in the object or the part (We provide some examples in Figure 21 in the revised version of the paper),
> Even when users provide manually annotated masks, our approach is still preferred over inpainting (see 'Mask-Based Editing' in Table 1), despite the inherent advantage that manually annotated masks offer to mask-based methods.
> This demonstrates the strength and the impact of our approach.
>
> W2 - Our approach optimizes a token per part, where each token is a vector with the dimensionality 2048x2 as explained in section 3.1. We only store those tokens on disk, where each part token is only 17 KB. In practice, we can store as many tokens as possible and load them when requested by the user through the special token identified <part-name>.
>
> W3 - Our approach does not rely on specific datasets or pre-trained models. As we showed in Appendix C, 5-10 images are sufficient to achieve good localization. These images can be manually annotated or from parts datasets such as PascalPart or PartImageNet.
> The 7 tokens that we experimented with in the paper cover most of the common objects such as humanoids, animals, vehicles, and chairs. If the user is interested in learning specialized parts that are not provided, annotating the parts and optimizing the token is a straightforward process.
> For annotating the part, online annotation tools such as “MakeSense” take around 20-30 seconds per image, which is 3-5 minutes for 10 images.
> Then we will provide the source code for optimizing new tokens and using them for editing upon the acceptance of the paper to facilitate learning new part tokens for the community.
>
>
> W4 - As we explained in W2, we do not finetune the model weights and keep the model frozen during token optimization. This means no LoRA weights or changes to the underlying model.
>
> W5 - We did our best to enhance the figure by adding a legend and increasing the font size in the revised version for better readability on paper.

---

> > ### Comment · Reviewer_kfJM · 2024-11-25
> > **rebuttal**
> >
> > I thank the reviewers for their response. Thanks for pointing out my misunderstanding in W2; that is clear now. (I was confused because line 245-249 referred to the layers L for optimization, but now understand that these layers are frozen and used to optimize the token).
> >
> > All reviewers (including me) are convinced that part-based image editing is desirable. I am also convinced that the proposed method works. However, to make it a ICLR paper, there needs to be a significant technological or scientific contribution (W1). At the moment for me the contribution is too small; we know how to learn part-based tokens and their attention maps can subsequently be used for text-based image editing. Could you shortly summarize the main technological/scientific contributions, directly citing the most relevant methods and emphasizing the main differences with these. Did you use any new insight not used by other methods.

---

> > > ### Author Response · Authors · 2024-11-27
> > >
> > > We thank the reviewer for their thoughtful feedback and engagement in the discussion.
> > >
> > > First, we would like to clarify that *token optimization* is a general technique, much like LoRA, which has been utilized across multiple *published* works to address different problems. Notable examples include concept learning in Avrahami et al., 2023a (SGA 2023), Safaee et al., 2024 (CVPR 2024), and unsupervised keypoint detection in Hedlin et al., 2023 (CVPR 2024). These works incorporated application-specific components to effectively integrate token optimization into their respective solutions. The effectiveness of token optimization for part-based image editing might appear self-evident in retrospect, but that is largely because we have successfully demonstrated its utility in this new domain.
> > >
> > > While determining whether our contributions are sufficient for acceptance at ICLR is inherently subjective, we believe our work offers significant value through *three* key types of contributions:
> > >
> > > **a) Conceptual Contribution:**
> > > We present the first approach for text-based fine-grained editing, which is more intuitive, user-friendly, and powerful compared to the traditional, less convenient mask-based editing methods.
> > >
> > > **b) Algorithmic Contributions:**
> > > We proposed a novel editing algorithm that integrates three diffusion paths, enabling fine-grained editing in a single inference pass using optimized part tokens.
> > > We also conducted a comprehensive analysis of several core aspects, including training and inference timesteps, layer selection, data scalability, and token padding strategies.
> > > Our approach is designed to be dynamic, and we successfully adapted it for different Stable Diffusion models (SDXL and SD 1.5/2.1), thereby providing the community with deeper insights into the potential of token optimization for image editing across different architectures.
> > >
> > > Furthermore, we proposed a novel mask computation algorithm that generates non-binary editing masks, utilizing an adaptive thresholding strategy to produce seamless, natural edits.  Results generated by this algorithm even outperformed *mask-based* editing approaches, where manually annotated binary masks were provided, as shown in Table 1. This novel strategy sets a precedent for future advancements in image editing.
> > >
> > > **c) Strong and Exhaustive Results:**
> > > We conducted extensive experiments and comparisons that demonstrated the effectiveness of our approach in various settings. Additionally, we highlighted the limitations of current state-of-the-art editing approaches, particularly in the context of fine-grained editing and racial bias (Figure 5). These insights pave the way for future research to address these challenges.
> > >
> > > While we understand that some might view contribution (a) alone as insufficient, we believe the combined package of contributions (a), (b), and (c) provides substantial value to merit acceptance.

---

> > > > ### Comment · Reviewer_kfJM · 2024-11-28
> > > >
> > > > Thank you for your answer. I remain unconvinced that the technological or scientific contribution merits an ICLR paper. You write that the 'he effectiveness of token optimization for part-based image editing might appear self-evident in retrospect, but that is largely because we have successfully demonstrated its utility in this new domain.', but I am not surprised that it works for parts or actually any other semantic localizable description of images (like objects, parts, but also adjectives on color or texture etc). But this part, boils down to applying an existing methods to new data (part-data). I appreciate the inference timestep and layer selection but also consider these minor contributions. Given these considerations, I remain with my rating.

---

### Official Review · Reviewer_rJwB · 2024-10-29

**Soundness:** 3
**Presentation:** 3
**Contribution:** 2
**Rating:** 5
**Confidence:** 4

**Summary:**

This paper introduces a text-based image editing method for object parts using pre-trained diffusion models. It enhances model understanding of object parts for fine-grained edits through optimized textual tokens and masks.

**Strengths:**

1. The paper presents a flexible method for text-based image editing focused on object parts, which is a novel contribution to the field of image processing and editing.
2. The paper is well-written and easy to follow.
3. The authors have conducted extensive experiments and provide a solid basis for its practical application.

**Weaknesses:**

1. The approach relies on a finite and manually defined set of part tokens, which could restrict the flexibility and applicability of the method in real-world scenarios where users might need to edit object parts that are not covered by the predefined tokens. This limitation could affect the generalizability of the technique to a broader range of editing tasks and objects.
2. There are many methods nowadays that utilize semantic segmentation to create masks, which are quite similar to this paper. You should supplement your study with some relevant ablation experiments, like replace the attention mask with semantic segmentation part  and compare it with similar methods [1][2][3].

[1] SmartMask: Context Aware High-Fidelity Mask Generation for Fine-grained Object Insertion and Layout Control.
[2] Accelerating Text-to-Image Editing via Cache-Enabled Sparse Diffusion Inference.
[3] Towards Understanding Cross and Self-Attention in Stable Diffusion for Text-Guided Image Editing.

**Questions:**

1. The paper does not explicitly mention how it deals with fine-grained edits for multiple objects within an image, such as distinguishing between two heads in an image for editing purposes. could you provide some form of a mechanism to differentiate between objects? How your method deal with this situation?
2. How many part tokens can you serve?
3. What's your random methods for initializing textual embeddings?
4. How do you generate reliable localization masks? Does this process rely on specific datasets or pre-trained models? Will the distribution of the training data for the mask overlap with the distribution of the images used for testing?
5. see weakness.

---

> ### Author Response · Authors · 2024-11-24
>
> W1 - We completely understand this concern, but in practice, the 7 tokens that we experimented with in the paper cover most of the common objects such as humanoids, animals, vehicles, and chairs. If the user is interested in learning specialized parts that are not provided, annotating the parts and optimizing the token is a straightforward process.
> For annotating the part, online annotation tools such as “MakeSense” take around 20-30 seconds per image, which is around 3-5 minutes for 10 images.
> Then, we will provide the source code for optimizing new tokens and using them for editing upon the acceptance of the paper to facilitate learning new part tokens for the community.
>
> W2 - We provided these experiments in Table 1 under “Mask-Based Editing.” In these experiments, we replaced the attention with manually annotated segmentation masks to act as an upper bound. Despite that, providing the masks gives a clear advantage to these methods over our approach, which is only text-based. Our approach was favored by users in the user study.
>
> Q1 - If an image has two objects of the same scale with the same part that is being edited, the edit will most likely apply to both objects. A potential mechanism for choosing which object to edit is optimizing tokens for <left>, <center>, and <right> that are combined with the part tokens. However, this approach does not scale to more than 3 objects. Another approach is using a Vision Language Model to parse the editing prompt into a localization mask to mask only the object of interest. We leave these investigations for future work.
>
> Q2 - Our approach can serve as many parts as possible since each optimized token is saved to disk (17 KB) and can be loaded upon request (when the user includes <part-token> in the editing prompt).
>
> Q3 - We use random normal initialization.
>
> Q4 - Our approach does not rely on specific datasets or pre-trained models. As we showed in Appendix C, 5-10 images are sufficient to achieve good localization. These images can be manually annotated or from parts datasets such as PascalPart or PartImageNet.
> We ensure that the training and the test set do not overlap.
> To demonstrate that our approach does not depend on the choice of training images and that the training and test set do not overlap, we conduct a 5-fold cross-validation experiment on the “<head>” token from PartImageNet.
> We obtain an average mIoU of 71.704 and a standard deviation of 4.372. These results demonstrate that our approach performs consistently well, irrespective of the choice of the training images.
> This is a consequence of the semantically rich features of pre-trained diffusion models highlighted in “Emergent Correspondence from Image Diffusion, Neurip 2024”.

---

> ### Author Response · Authors · 2024-12-04
>
> We believe that we have adequately addressed your concerns, as no follow-up questions or further discussions were raised. We respectfully request the reviewer to reconsider their rating if the additional information and rebuttal sufficiently resolve the issues highlighted.

---

### Official Review · Reviewer_8GnP · 2024-11-03

**Soundness:** 3
**Presentation:** 4
**Contribution:** 3
**Rating:** 6
**Confidence:** 4

**Summary:**

The paper introduces a method to enhance pre-trained diffusion models for fine-grained image editing by training part-specific tokens for localizing edits at each denoising step. This approach uses feature blending and adaptive thresholding for seamless edits while preserving unaltered areas. A token optimization process expands the model’s semantic understanding without retraining, using existing or user-provided datasets. Qualitative as well as quantitative experimental comparison have been conducted to demonstrate the effect of the proposed method.

**Strengths:**

1. This paper addresses a critical problem in image editing: the inability to accurately edit specific parts of an object while keeping the rest of the image unchanged.

2. The use of token optimization to learn adaptable tokens for subsequent editing tasks is intuitive and intriguing.

3. The experiments are thorough, with comprehensive ablation studies that validate the effectiveness of the proposed approach.

4. The paper is well-written, easy to follow, and logically structured.

**Weaknesses:**

1. The images used for training part tokens are very limited, with only 10–20 images. In such cases, the representativeness of the images is crucial for generalization. It would strengthen the paper if the authors would conduct experiments to show the impact of varying the types of training images on the model's performance.

2. The method involves many hyperparameters that require tuning, including the number of diffusion timesteps for training part tokens and inference, the selection of layers for optimization, and the adjustable tolerance for transitions between edited parts and the original object. This adds complexity to the overall framework and could make it challenging to implement effectively.

3. In practical scenarios, one might want to adjust two parts simultaneously. Therefore, how will the method apply when handling instruction text that requires simultaneous editing of two parts? I suggest the authors include experiments or examples to show the model's performance on multi-part edits.

4. Will the evaluation dataset for PartEdit be made publicly available? Also, will the code be available?

5. Typo: The text inside Figure 1, "prompt-to-prompt (ICRL2023)," should be corrected to "ICLR."

**Questions:**

1. Given the limited number (10–20) of images used for training part tokens, how were these images selected to ensure representativeness, and what impact does this selection have on the model's generalization capabilities?

2. Are there guidelines or best practices provided for hyperparameter tuning?

3. How effectively does the method handle instructions that require simultaneous modifications to two or more parts of an image?

---

> ### Author Response · Authors · 2024-11-24
>
> Q1/W1 - To demonstrate that our approach does not depend on the choice of training images, we conduct a 5-fold cross-validation experiment on the “<head>” token from PartImageNet.
> We obtain an average mIoU of 71.704 and a standard deviation of 4.372. These results demonstrate that our approach performs consistently well, irrespective of the choice of the training images.
> This is a consequence of the semantically rich features of pre-trained diffusion models highlighted in “Emergent Correspondence from Image Diffusion, Neurip 2024”. We expanded the supplementary material under "impact of choice of images for training part tokens".
>
> Q2/W2 - We understand the concern, and for this reason, we added the Hyperparameters section to Appendix L (in addition to the discussion of some parameters in Figure 10 - 12).
> We are also releasing the code, the evaluation benchmarks, and a demo upon acceptance of the paper to facilitate future development.
>
> Q3/W3 - Our approach is versatile and can incorporate multiple part edits simultaneously. We provide some examples in Figure 19 in the appendix of the revised version of the paper (please see the revised PDF). In this setting, when the user specifies multiple tokens in the editing prompt, the tokens are loaded and fed through the network to compute cross-attention maps per part token.
> Then, we accumulate these maps across layers and normalize them jointly across different parts. We provide visualizations for the combined blending masks in Figure 19.
>
> W4 - Our source code for optimizing the tokens, editing, and evaluation datasets will be made publicly available upon the acceptance of the paper.
>
> W5 - We fixed the typo in the updated version of the paper.

---

> > ### Comment · Reviewer_8GnP · 2024-11-26
> >
> > The authors have addressed primary concerns raised in the initial review. They conducted a 5-fold cross-validation experiment to demonstrate the robustness of their approach to the choice of training images, which is a significant improvement. The commitment to release the code and evaluation benchmarks upon acceptance are also positive steps. However, I still have concerns about the complexity of the method and the ability to handle multiple part edit simultaneously. Hence,  I maintain my original recommendation.

---

> > > ### Author Response · Authors · 2024-11-27
> > >
> > > We're glad that we were able to address your primary concerns.
> > >
> > > Regarding the complexity of the method, we analyze the impact of different hyperparameters in the paper to optimize them for fine-grained image editing.
> > > However, all suggested hyperparameters are *stable*, and the user does not need to tune any of them to obtain good results.
> > > We only allow the user to *optionally* control the parameter $t_e$ to control the locality of the edit.
> > > We have included a screenshot of our Gradio user interface in the revised version of the paper (Appendix Q and Figure 22), which demonstrates how easy it is to use our method in practice with no parameter tuning at all.
> > >
> > > Regarding handling multiple parts, our pipeline is quite flexible in that term. To edit multiple parts, the user would either edit different parts sequentially or set the editing prompt in the form “with a <edit> <part-1> <part-2>”
> > > These are largely implementation details, and we welcome suggestions on how to make this process more efficient and user-friendly.

---

### Official Review · Reviewer_SEmy · 2024-11-04

**Soundness:** 3
**Presentation:** 2
**Contribution:** 2
**Rating:** 5
**Confidence:** 4

**Summary:**

This paper proposes an inference-based image editing method that can perform fine-grained object parts editing. Specifically, this paper trains part-specific tokens that specialize in localizing the editing region at each denoising step, then develop feature blending and adaptive thresholding strategies that ensure editing while preserving the unedited areas.

**Strengths:**

(1) This paper focus on an interesting question, which as great significance to downstreaming research and tasks.

(2) The overall design of the model design is generally make sense.

(3) This paper is easy to follow.

**Weaknesses:**

(1) Some related works are missing. Discussing and compare these related works will be good for improving the paper's quality.

- [1] Pnp inversion: Boosting diffusion-based editing with 3 lines of code

- [2] Inversion-free image editing with natural language

- [3] Dragondiffusion: Enabling drag-style manipulation on diffusion models

(2) Can you provide more experimental results to prove the effectiveness of the proposed method? For example, more comparison results with training-based editing methods such as InstructPix2Pix[4]. More visualization for editing regions of various image-editing prompt pairs. Results of combining the proposed method to different pretrain checkpoints/different diffusion model backbones to show its generalization ability.

- [4] Instructpix2pix: Learning to follow image editing instructions

**Questions:**

See weakness.

---

> ### Author Response · Authors · 2024-11-24
>
> W1 - As per the reviewer’s request, we expand Table 1 in the main paper with quantitative comparisons against PnPInversion [1] and InfEdit [2] and qualitative comparisons in Figure 18 in the supplementary material.
> Our approach outperforms both methods by a huge margin and is favored by users 79.8% and 77.8% of the time, respectively.
> For DragonDiffusion, we find that it is a drag-based editing method and, therefore, not directly comparable to our approach. However, we include it in the related work section.
>
> W2 - **InstructPix2Pix Comparison** Table 1 of the main paper already has InstructPix2PIx [4] under “iP2P” as described in Evaluation setup section 4.1. We use the same shortened abbreviations in Figure 5. The results show that our approach is preferred by 77% of the users over iP2P. Figure 5 also shows that iP2P consistently fails at identifying parts and edits the whole object instead.
>
> **Visualization of the editing regions** We provide an additional Figure in the appendix of the revised version of the paper (Figure 20) that shows visualizations of the editing mask for different editing regions (Please see the revised PDF). The figure shows the continuous nature of our blending masks compared to the conventional binary masks used for inpainting.
>
> **Applicability to Different Diffusion Models** Our approach can be applied to any UNet-based diffusion model, i.e., any version of the Stable Diffusion family of models. In the paper, we use SDXL for the synthetic image setup, while we use SD 2.1 for the real image editing as we employ Leedits++ as a baseline for this setting.
> Our approach can also be applied to SD 1.5 as it has the same architecture as SD 2.1.

---

> ### Author Response · Authors · 2024-12-04
>
> We believe that we have adequately addressed your concerns, as no follow-up questions or further discussions were raised. We respectfully request the reviewer to reconsider their rating if the additional information and rebuttal sufficiently resolve the issues highlighted.

---

### Author Response · Authors · 2024-11-24

We thank the reviewer for all the comments, positive feedback, and questions. We address the concerns of reviewers individually for each question/weakness but encourage others to read the responses of other reviewers. Most reviewers see our method in a positive light, including that we address a critical problem in image editing (8GnP), an interesting question with great significance for downstream tasks and research (SEmy). The paper is easy to follow (SEmy, 8GnP, rJwB) and provides extensive experiments and practical applications (rJwB, 8GnP). The most common question was about the number of images (10-20), which we addressed by adding extra cross-validation experiments (“Impact of Images for training part tokens” in supplementary). Additionally, we want to note that we are aware that training with more images improves localization, but rather, we see it as a strength that our method works with limited images that can be manually annotated or annotated using some other binary mask methods. We added extra experiments and user study results in Table 1 (SEmy), extra works in related sections (SEmy, rJwB), and extra sections in supplementary sections for questions from the reviewers. We have addressed kfJM's concerns, which may have stemmed from the print version of Figure 3. To clarify that only tokens are trained, we have updated the figure by increasing the text size and adding a legend. Specifically, our approach focuses solely on training tokens and does not involve handling multiple models or LoRAs, as mentioned in W2/W4.
We sincerely welcome the reviewers to reconsider their evaluation and update their score if they feel our clarifications and revisions have strengthened the submission. We hope our responses address your concerns, but if there are any remaining questions or additional points to clarify, please feel free to post them.

---

### Meta-Review · Area_Chair_ub6d · 2024-12-17

**Metareview:**

This paper presents an image editing method that can perform fine-grained object parts editing. This paper was reviewed by 4 experts in the field, and received 3, 5, 5, 6 scores. The reviewers find that this paper is well-written and easy to follow. However, some critical issues are pointed out in the reviews: insufficient technical contributions, complexity of the method, insufficient experiments. The rebuttal does not fully address these concerns. After rebuttal, this paper receives 3 negative ratings in the final rating, while Reviewer 8GnP though gave positive rating still have concern on complexity of the method. The AC recommends rejection mainly due to its limited technical contribution. The authors are encouraged to consider the reviewers' comments when revising the paper for submission elsewhere.

**Additional Comments On Reviewer Discussion:**

This paper was reviewed by 4 experts in the field, and received 3, 5, 5, 6 scores. Reviewer kfJM keeps the initial rating 3 after reading the authors' rebuttal, because of the concerns on insufficient technological or scientific contribution. Reviewer 8GnP gave the only positive rating, but still had concerns about the complexity of the method and the ability to handle multiple part edit simultaneously on discussion period. After reading the paper and the authors' rebuttal, the AC agrees with the Reviewers concerns, especially, the limited technical contribution.

---

### Decision · Program_Chairs · 2025-01-22

Reject